# Optimizing solar drying efficiency: Effects of PCM, and IR on energy and exergy performance

**Mehdi Moradi** *, Reza Raeesi, Dariush Zare, Mahdi Keramat-Jahromi

Biosystems Engineering Department, School of Agriculture, Shiraz University, Shiraz, Iran

* moradih@shirazu.ac.ir; mehdimoradi.ir@gmail.com (MM)

## Abstract

This study investigates the drying behavior of potatoes using a hybrid solar dryer equipped with a Compound Parabolic Concentrator (CPC) collector, Phase Change Materials (PCM), and Infrared Radiation (IR). Drying experiments were conducted at 40°C, 50°C, and 60°C under different PCM and IR configurations to evaluate drying kinetics, energy consumption, and product color quality. Energy and exergy analyses, along with assessments of drying time and color change (ΔE), were performed to identify the most efficient drying conditions. This study introduces a novel integration of PCM and IR in a hybrid solar drying system, providing a unique approach to optimizing energy efficiency, and product quality. The results demonstrated that PCM significantly improved the drying process by reducing drying time by an average of 5.3%, stabilizing the thermal environment, and enhancing both energy and exergy efficiency. The lowest Specific Energy Consumption (SEC) and shortest drying time were recorded at 60°C with PCM and IR, demonstrating the efficiency of this setup in reducing energy consumption while ensuring high drying performance. IR alone reduced drying time by 40%, accelerating moisture removal considerably. However, while the combination of PCM and IR enhanced thermal stability, it slightly prolonged drying time due to PCM's heat absorption characteristics. Among the tested conditions, 60°C with PCM and IR was identified as the optimal setting, achieving the lowest SEC while minimizing drying time and color degradation. This study highlights the importance of integrating PCM and IR into solar drying systems to enhance efficiency, reduce energy consumption, and improve product quality. Future research should explore additional drying techniques, such as microwave and ultrasound-assisted drying, to further optimize hybrid solar drying systems.

## Introduction

In recent years, global attention has increasingly focused on energy production and efficiency due to population growth, rising food consumption, and the pressing need to reduce dependence on fossil fuels. These trends have driven growing interest in

**Data availability statement:** "All relevant data are within the paper and its Supporting Information files."

**Funding:** The author(s) received no specific funding for this work.

**Competing interests:** he authors have declared that no competing interests exist.

renewable energy sources, particularly within the agricultural sector, where energy consumption plays a significant role. For example, approximately 30% of energy used in agriculture is dedicated to processing, with drying operations accounting for about 62.3% of this energy expenditure [1]. Freshly harvested agricultural products often have high moisture content, which presents substantial storage challenges, including the growth of bacteria and fungi. To mitigate these issues and preserve product color quality, effective drying methods are essential. Traditional open-air sun-drying, while common, is associated with long drying times and an increased risk of contamination, which can compromise the product's quality. To address these drawbacks, various types of solar dryers have been developed in recent years [2,3]. Industrial dryers, equipped with advanced control systems, are designed to monitor and regulate key drying parameters such as airflow rate and inlet air temperature. However, poor control of these parameters can still lead to suboptimal drying performance, wasting energy, and reducing efficiency [4].

Recent innovations have aimed at improving the efficiency of drying systems. For instance, Rezaei et al. [5] demonstrated that integrating a coil absorption plate with Phase Change Materials (PCM) in the dryer enhanced both the collector and dryer efficiency by 28.5% and 52.1%, respectively, compared to using only the plate. Similarly, Mandal et al. [6] conducted experiments on a hybrid solar dryer for drying mint and coriander, where paraffin wax served as the PCM. In this experiment, increasing the bed depth resulted in an improved drying rate. IR heating has garnered significant attention across various industries for its potential to improve efficiency and performance, as noted by Liu et al. [7] and Hao et al. [8]. In another study, the combination of infrared (IR) radiation and PCM was found to significantly increase the drying rate of pineapple in an IR-solar hybrid dryer [9].

Thermodynamic analysis, particularly energy and exergy analysis, plays a crucial role in the design and optimization of thermal systems. While energy analysis focuses on the conservation and transfer of energy, exergy analysis provides deeper insights by quantifying the system's potential to perform useful work when interacting with a heat reservoir. Exergy analysis explicitly accounts for irreversibilities within a process, such as entropy generation and energy degradation. Unlike energy, which is conserved and merely changes form, exergy emphasizes the quality and usability of energy, offering a valuable metric for identifying and reducing inefficiencies. Destruction of exergy is directly proportional to the creation of entropy within the system and its surroundings, highlighting the importance of exergy analysis in improving energy efficiency in industrial applications. By identifying and addressing sources of irreversibility, this approach facilitates the design of more efficient and sustainable thermal processes. A study by Ghasemi et al. [10] investigated the effects of infrared (IR) heating at various temperatures (40°C, 50°C, and 60°C) on energy and exergy efficiency. The results indicated that IR heating was most effective at 40°C, where Specific Energy Consumption (SEC) decreased by 12.18% and the Energy Utilization Ratio (EUR) increased by 14.4%. However, the benefits of IR heating diminished at higher temperatures. Hybrid dryers performed optimally at 40°C, while conventional solar dryers showed superior exergy efficiency at 50°C and 60°C. These findings

have contributed to the development of an optimized criterion for utilizing IR in hybrid drying systems. To evaluate the impact of key parameters on drying kinetics and energy performance, this study examines a hybrid solar dryer equipped with a Compound Parabolic Concentrator (CPC) collector. Previously used by Ebadi et al. [11] to assess the energy and exergy efficiency of tomato drying, this dryer has been enhanced with the integration of PCM and IR technologies. The combination of these advanced thermal control methods offers an innovative approach to optimizing drying processes. This research employs a comprehensive multi-objective evaluation framework to assess energy efficiency, exergy performance, drying time, and product color quality. The findings contribute to the advancement of renewable energy applications in food processing and provide scalable solutions for improving sustainability in agricultural drying. The innovations presented here lay the groundwork for the development of intelligent, adaptable, and energy-efficient drying systems that can be applied to a wide range of agricultural and biological products.

## Materials and methods

### Sample preparation

Fresh potato samples with an initial moisture content of 83±2% (wet basis) were procured daily from a local market to ensure freshness. After being thoroughly washed and peeled, the potatoes were sliced into 60 g samples (Fig 1), each with a thickness of 1 mm and approximately uniform diameters, using a slicer. The initial mass of the product is measured using an A&D scale manufactured in Japan, with a precision of 0.001 g. To determine the initial moisture content of the samples, an electric oven (model BMS55, manufactured by Fan Azma Gostar, Iran) is used at a temperature of 105 °C for 24 hours.

### Solar dryer

The solar dryer utilized in this study consists of several key components: a compound parabolic collector (2), an air temperature regulator (3), an equalizing chamber (4), a fan (5), a heating channel (6), a diffuser (7), and a drying chamber (8) (Fig 2).

The dryer's thermal energy is primarily supplied by solar radiation absorbed by the collector's surface. When solar radiation decreases, a 1.5 kW electric-powered heating channel is activated to maintain performance. The collector is a non-vacuum compound parabolic type with an effective area of 2.4 m² and a concentration ratio of 2.5, designed to efficiently heat the air. The absorber tubes, positioned at the focal point of the parabolic collectors, are painted black to maximize light absorption. Additionally, 0.4 cm thick stainless-steel reflector panels are used to concentrate sunlight onto the absorber tubes, enhancing heat transfer. The collector's installation angle relative to the horizontal plane (α) is determined using Equation (1) [12]:

$$\alpha = 15 + \beta \tag{1}$$

Where β represents the latitude of the location.

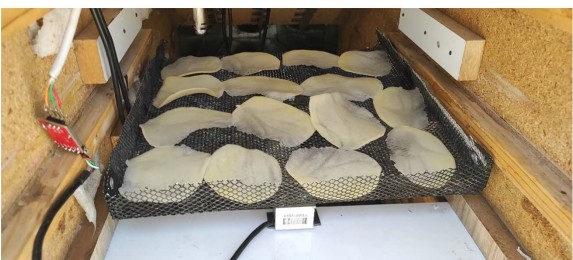

**Fig 1. Samples inside the drying cabinet.**

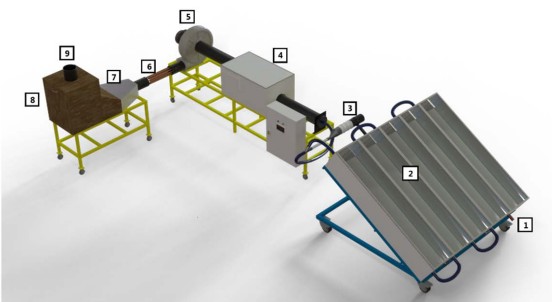

**Fig 2. Schematic of dryer system.** 1- Inlet of ambient air, 2- CPC solar collector, 3- Air temperature regulator, 4- Equalizing chamber, 5- Fan, 6- Auxiliary heater, 7- Diffusion chamber, 8- Drying chamber, 9- Air outlet.

The experiments were conducted in the Biosystems Engineering Department at Shiraz University, located in the Bajgah region at approximately 30° latitude. Accordingly, the collector installation angle (α) was set to 45°. To prevent excessive air temperatures inside the drying system, an electronic damper was installed for temperature regulation. Positioned between the collector and the entrance to the equalizing chamber, the damper is controlled by a stepper motor (model Ts310n247) with a torque of 6.5 kg. Programmed using Arduino software, the system automatically opens the damper to introduce ambient air when the drying chamber temperature exceeds the preset threshold, effectively lowering the temperature (Fig 3). An air blower, powered by a single-phase 0.4 kW electric motor from Techtop (Italy), ensures the required airflow throughout the system. Located before the air blower, the equalizing chamber stabilizes the heated air flow. This rectangular chamber, measuring 30 × 50 × 70 cm, is constructed from particle board. The drying chamber, positioned after the heater, contains a drying tray and an IR lamp (Fig 4). The distance between the IR lamp and the product tray is set at 30 cm to ensure optimal drying conditions. Inside the drying chamber, a mesh tray made of galvanized sheet metal (25 × 25 cm) holds the drying product, facilitating uniform heat distribution.

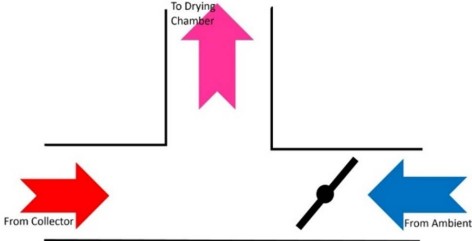

**Fig 3. Air temperature regulator.**

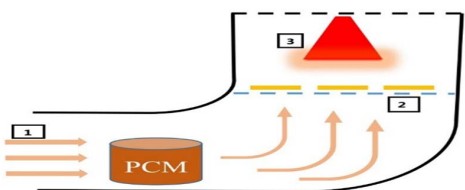

**Fig 4. Schematic of the drying cabinet.** 1- Air inlet to the drying cabinet 2- Product tray 3- IR lamp.

A force gauge (model L6D, Class C3, with a capacity of 3 kg) continuously monitors the product's mass. The drying process is considered complete when the product's moisture content reaches approximately 10% (dry basis), as indicated by the control settings. With the initial moisture content established, any reduction in the product's mass (recorded by the load cell) corresponds to water loss, enabling real-time calculation of the product's moisture content throughout the drying process.

For the drying experiments involving PCM, 1500 g of paraffin was obtained, melted, and poured into small aluminum containers, each holding 150 g. During the experiments, these containers were placed at the bottom of the drying cabinet, positioned before the airflow reached the product.

## Temperature Controller

To stabilize the temperature within the drying chamber, a temperature controller is used. This device regulates the temperature automatically, without requiring staff intervention, ensuring precise control. It compares the actual temperature with the set value and adjusts the control element accordingly (Fig 5). Specifically, when the actual temperature is below the set value, the temperature controller activates the heating element. Once the set temperature is reached, the controller turns off the heating element.

In this study, a temperature controller (model XMT-803, manufactured in China) was used to regulate the drying process. A solid-state relay (SSR) model 25DA (produced by CRYDOM) and a PT100 temperature sensor with an accuracy of ±0.1°C were installed at the drying chamber's outlet. Ten PT100 sensors were strategically placed throughout the system to monitor temperature distribution. To measure relative humidity before and after the drying chamber, an SHT15 temperature and humidity sensor was employed, offering a humidity accuracy of ±0.05% RH and a temperature accuracy of ±0.1°C. The air mass flow rate was measured at the drying chamber inlet using a hot-wire anemometer (model Testo 435) with an accuracy of ±0.03 m/s. The mass flow rate was then calculated based on air density and the cross-sectional area of airflow. In this study, the air velocity at the fan inlet (with a cross-sectional area of 182 cm²) was set to 2 m/s.

## Design of study

The experiments were conducted at three different air temperatures: 40, 50, and 60 °C, which are within the common range for potato drying as reported in the literature [13,14]. Each temperature condition was tested under two scenarios: one with PCM and one without. Additionally, two IR heating setups were used, one with a 250-watt IR lamp and one without. The experiments followed a factorial design with a completely randomized layout and were conducted in triplicate.

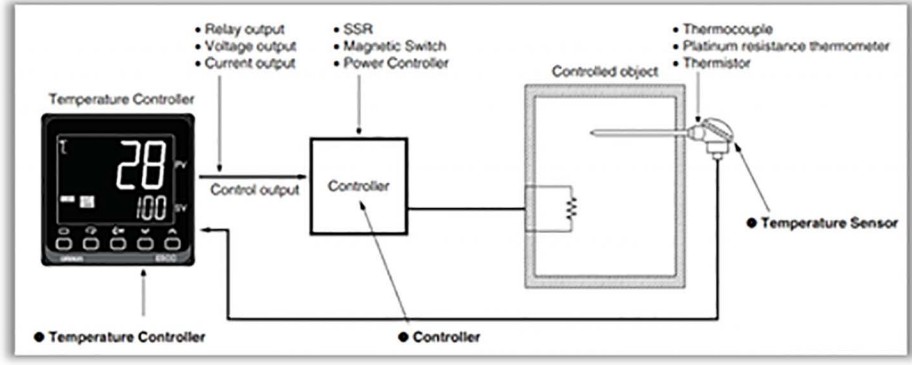

**Fig 5. Temperature control circuit.**

## Color Change analysis

To evaluate color variations, potato samples were placed in a custom-designed imaging box before and after the drying process. Images were captured using a Xiaomi 5G camera with a 64-megapixel wide-angle lens and analyzed in MATLAB R2018b. The Lab color space values were extracted for each sample. The total color change (ΔE) was then calculated using the extracted values (Equation 2) [15].

$$\Delta E = \sqrt{(\Delta L)^2 + (\Delta a)^2 + (\Delta b)^2}$$

(2)

## Energy and Exergy Analysis

**Energy.** The consumption of electrical energy is quantified using a digital counter that has an accuracy of 0.1 kilowatt-hours. Also, by multiplying the collector area into the solar irradiance, input solar energy ($E_s$) for product drying was calculated. The SEC of drying was calculated using (Equation 3) [16]:

$$SEC = \frac{E_e + E_{IR} + E_s}{M_w}$$

(3)

In this context, $E_e$ represents the electrical energy consumed during the drying process which includes contributions from the exhaust fan and electric heater (kJ), $E_{IR}$ denotes the energy consumed by the IR lamp (kJ). Additionally, $M_w$ refers to the mass of water evaporated from the product (kg). The EUR was derived from (Equation 4):

$$EUR = \frac{E_{out}}{E_{in}}$$

(4)

Where, $E_{out}$, and $E_{in}$ are useful and total input Energy, respectively.
The calculation of useful energy (the power consumed for evaporating moisture from the drying material) was performed using (Equation 5): [17]

$$E_{out} = m_s l_g t$$

(5)

Where:
$E_{out}$: Useful energy (kJ)
$m_s$ is the Rate of evaporation of the drying material (kg/s)
$l_g$: Latent heat of vaporization of water (kJ/kg)
$t$: Duration of the drying process (s)
**Calculation of Exergy Efficiency.** To determine the exergy efficiency, it is first necessary to calculate the input, output, and absorbed exergy. The input exergy to the drying chamber is calculated using (Equation 6) [18]:

$$Ex_{in} = \dot{m}c_p \left( T_{in} - T_a - T_{ambient} \ln \frac{T_{in}}{T_a} \right)$$

(6)

Where:
$Ex_{in}$: Input exergy rate (kW)
$\dot{m}$: Mass flow rate of the fluid (kg/s)
$T_{in}$: Inlet air temperature of the drying chamber (°C)

$T_a$: Ambient air temperature (°C)

Subsequently, the output exergy from the chamber is calculated using (Equation 7):

$$Ex_{out} = \dot{m}c_p(T_{out} - T_a - T_a \ln \frac{T_{out}}{T_a})$$

(7)

Where

$Ex_{out}$: Output exergy rate (kW)

$T_{out}$: Outlet air temperature of the drying chamber (°C)

The exergy of the IR source is then obtained using (Equation 8) [19]:

$$Ex_{IR} = \left( \varepsilon_{IR} + \frac{1}{3} \left( \frac{T_{ambient}}{T_{IR}} \right)^4 - \frac{4}{3} \varepsilon_{IR}^{\frac{3}{4}} \left( \frac{T_{ambient}}{T_{IR}} \right) \right) \times \sigma T_{IR}^4 A_{IR} \times t$$

(8)

Where:

$Ex_{IR}$: Rate of IR exergy consumption (kW)

$\varepsilon_{IR}$: Emissivity coefficient

$T_{IR}$: Surface temperature of the IR source (K)

σ: Stefan-Boltzmann constant

$A_{IR}$: Surface area of the IR source (m²)

Finally, the exergy efficiency and losses were determined using (Equations 9 and 10), respectively [10]:

$$\eta_{ex} = \frac{Ex_{out}}{Ex_{in} + Ex_{IR}}$$

(9)

$$Ex_{loss} = Ex_{IR} + Ex_{in} - Ex_{out}$$

(10)

Where:

$\eta_{ex}$: Exergy efficiency

$Exl_{oss}$: Exergy losses (kW)

$Ex_{in}$: Input exergy rate (kW)

$Ex_{out}$: Output exergy rate (kW)

$Ex_{IR}$: Rate of IR exergy consumption (kW)

## Data Analysis

The data collected from the experiments, were analyzed using SPSS 16 software, and mean comparisons were conducted with Duncan's test.

## Results and discussions

### Drying time: The Influence of temperature, PCM, IR, and their interactions

The analysis of variance results for drying time are presented in Table 1).

Accordingly, temperature ($p < 0.01$), IR ($p < 0.01$), PCM ($p < 0.05$), and the interaction between IR and PCM ($p < 0.01$) had significant effects on the drying time of potatoes. Among these factors, IR had the most substantial impact, as indicated by the highest F-value, while PCM had the least influence. Similarly, a study on the drying kinetics of green bell pepper slices in a fluidized bed dryer found that temperature significantly affected drying time, with higher temperatures

**Table 1. Analysis of variance for the effect of independent factors on drying time.**

| F | Mean square | Sum of squre | Df | Source |
|---|---|---|---|---|
| 89.61** | 3244.94 | 6489.88 | 2 | T |
| 456.01** | 16512.25 | 16512.25 | 1 | IR |
| 5.41* | 196.00 | 196.00 | 1 | PCM |
| 2.75ns | 99.44 | 198.88 | 2 | T × IR |
| 0.38 ns | 13.94 | 27.88 | 2 | T × PCM |
| 17.96** | 650.25 | 650.25 | 1 | IR × PCM |
| 0.50 ns | 18.19 | 36.38 | 2 | T × IR × PCM |
| – | 36.21 | 869.00 | 24 | Error |
| | – | 24980.52 | 35 | Total |

**: Significant in level of 1%

*: Significant in level of 5%

ns: non- significant

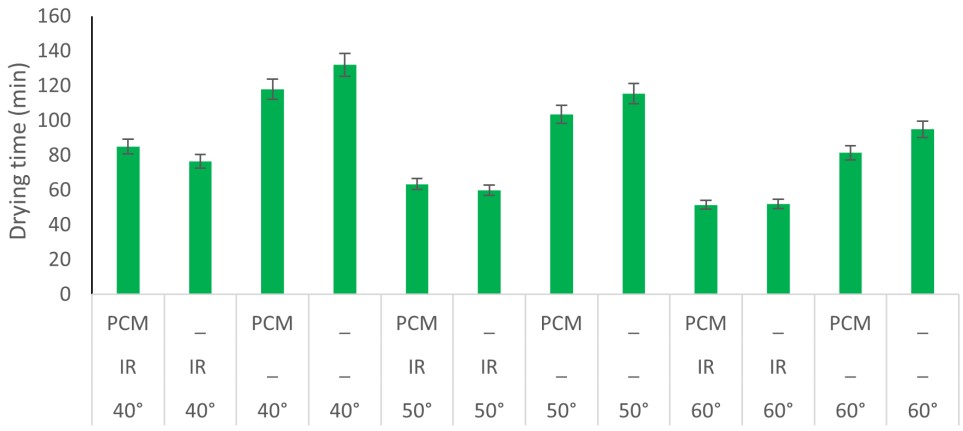

**Fig 6. Drying times under various drying conditions.**

leading to shorter drying durations [20]. Fig 6 compares the average drying time of potatoes under different experimental conditions. The longest drying time was observed at 40°C without IR and PCM, while the shortest drying time occurred at 60°C with both IR and PCM applied. The effects of various factors on drying time are detailed as follows:

### Effect of Temperature

The results indicate that higher temperatures lead to shorter drying times. Specifically, increasing the temperature from 40°C to 50°C reduced the drying time by an average of 16.8%, while a further increase from 50°C to 60°C decreased drying time by an average of 18.2%. Similarly, a study on the convective drying of potato slices found that drying time decreased with increasing temperature [13]. Another study on the drying kinetics of mint leaves also reported that higher temperatures resulted in shorter drying durations [21].

### Effect of IR

The average drying time for potato samples across all treatments using IR was 64.75 minutes, whereas treatments without IR had an average drying time of 107.58 minutes. This indicates that the implementation of IR technology resulted

in an average reduction of 40% in drying time. Similarly, a study on the drying of apple slices using IR reported a 50% reduction in drying time [22]. Another study comparing the drying process of peppermint (*Mentha piperita L.*) found that IR application could decrease drying time by up to 64% [23].

### Effect of PCM

The average drying time for samples treated with PCM was approximately 83.90 minutes, whereas samples without PCM had an average drying time of 88.50 minutes. This indicates that incorporating PCM resulted in an average drying time reduction of about 5.3%. However, the effect of PCM varied depending on IR conditions, which will be further discussed in the interaction effects section. Similarly, a study on the drying of jujube found that using PCM reduced drying time by 16.64% [24].

### Interaction Effect of IR and PCM

The average drying time with PCM under IR and non-IR conditions was 65.5 minutes and 108.83 minutes, respectively. In contrast, without PCM, the drying times under IR and non-IR conditions were 63.16 minutes and 119.33 minutes, respectively. These results confirm that IR technology significantly reduces drying time across all conditions. However, when IR is used in combination with PCM, drying time increases slightly. This is likely due to PCM's ability to absorb some of the IR heat, which may slow down the direct heating effect on the product.

### Drying kinetics

The drying kinetics of potato slices at three different temperatures are illustrated in Figs 7 to 9. Under all temperature conditions, the drying curves for experiments using PCM without IR exhibit the lowest slope during the initial drying stages. However, in the later stages, the slope increases and becomes steeper compared to the curve without PCM. This behavior suggests that, during the initial drying phase, PCM absorbs thermal energy from the hot air, delaying heat transfer to the product. As the drying process progresses, PCM undergoes a phase change and releases the stored energy, accelerating moisture removal in the later stages. As a result, this energy release significantly reduces the overall drying time compared to the control treatment without PCM. At 40°C, the most pronounced variation in moisture content is observed

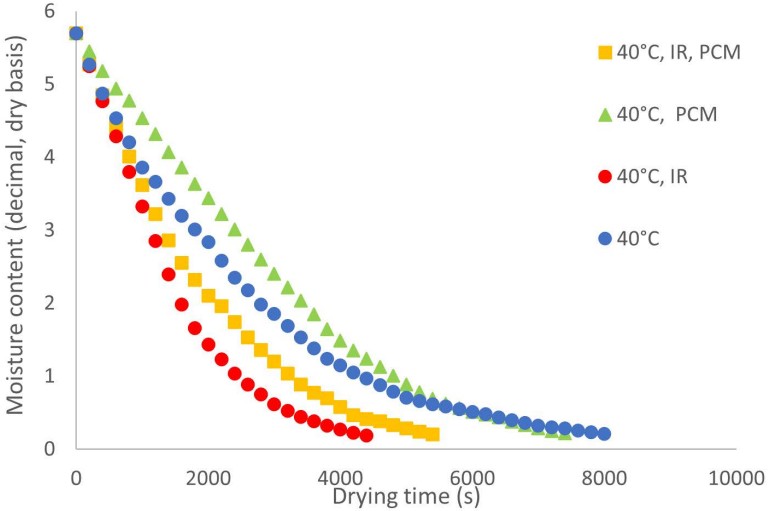

**Fig 7. Moisture content of potato samples during drying at 40 °C.**

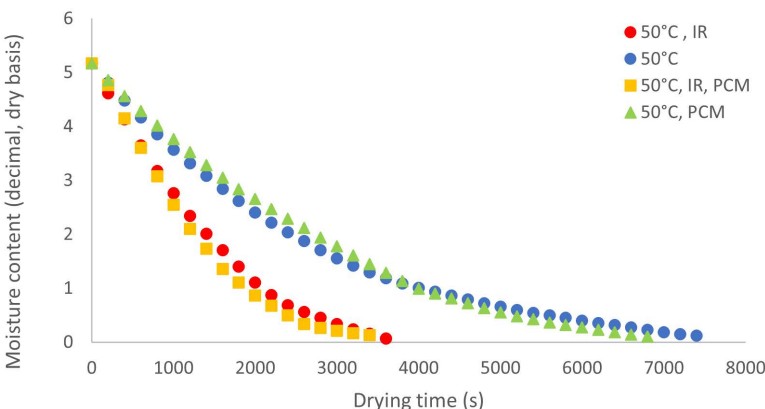

**Fig 8. Moisture content of potato samples during drying at 50 °C.**

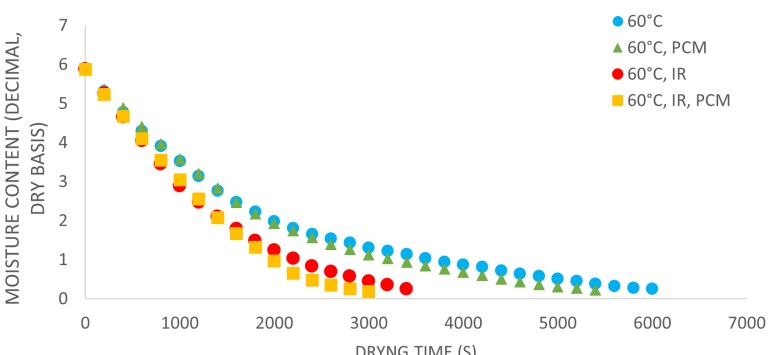

**Fig 9. Moisture content of potato samples during drying at 60 °C.**

in the treatment using infrared (IR) without PCM. From Figs 8 and 9, it is evident that the lowest slope of the moisture content curve occurs in the absence of both IR and PCM, while the steepest slope is seen in the treatment combining IR and PCM. It is important to note that the drying apparatus is designed to ensure that the drying air reaches the preset temperature (e.g., 40°C) before entering the product bed. However, the use of IR may cause the product bed temperature to exceed the intended level, further accelerating the drying process.

In another study that evaluated the effectiveness of solar drying and the energy efficiency of a dryer equipped with PCM and a recirculation system, the results showed that the simultaneous use of PCM and the recirculation system enhanced the thermal efficiency of the solar collector by 12.19% [24]. Furthermore, another study explored the impact of PCM on a solar collector specifically designed for drying pistachios. The research revealed that the incorporation of PCM significantly prolonged the operational period of the solar dryer, allowing it to function for up to 2 additional hours after sunset [25].

## Color Change Analysis

The impact of drying conditions—temperature, PCM, and IR—on the color change (ΔE) of potato slices is illustrated in Fig 10. Variations in ΔE result from chemical reactions, such as Maillard browning, as well as structural changes occurring during drying. The least color change was observed at 60 °C with PCM and IR, while the most significant change

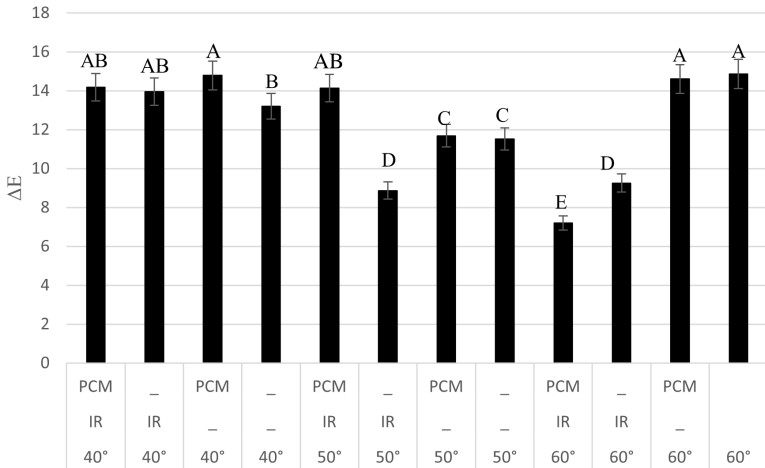

**Fig 10. Color changes of dried potato under different drying conditions.**

occurred at 40 °C with the same conditions. The average ΔE values at 40, 50, and 60 °C were 14.03, 11.55, and 11.48, respectively, indicating that the least color change occurred at 60 °C, likely due to a shorter drying time. The average ΔE for experiments with and without PCM was 12.77 and 11.95, respectively, while for experiments with and without IR, it was 11.27 and 13.45, respectively. These results suggest that higher temperatures and the presence of IR contribute positively to color preservation. However, the effect of PCM on color change varies, showing a beneficial influence at higher temperatures.

## Temperature regulator

To evaluate the impact of the temperature regulator (schematically shown in Fig 3), drying experiments were conducted under two conditions: with and without the regulator (Fig 11). The results indicate that implementing the temperature regulator significantly improved alignment between the drying chamber's temperature and the desired set point. Table 2) presents the discrepancy between the dryer's internal air temperature and the adjusted target

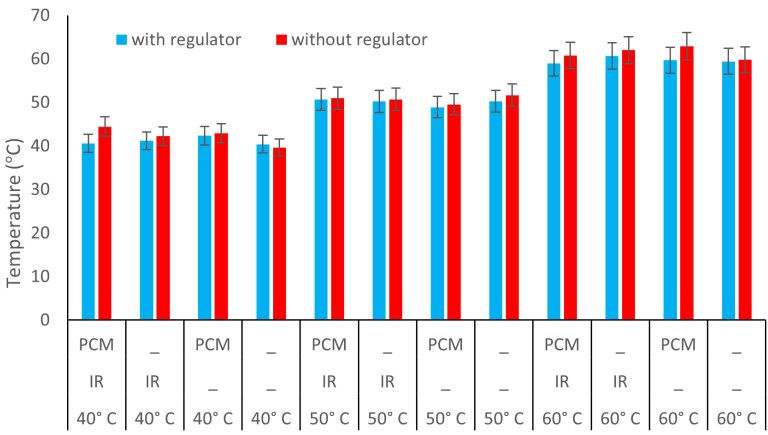

**Fig 11. Average temperature of drying chamber during the use and non-use of temperature regulator.**

**Table 2. The percentage difference between the actual temperature of the drying chamber and the set temperature in two scenarios: with a regulator versus without a regulator.**

| Drying Condition | Percentage of Temperature Difference between Actual and Desired (with Regulator) | Percentage of Temperature Difference between Actual and Desired (without Regulator) |
|---|---|---|
| 40-PCM-IR | 1.47 | 11.13 |
| 40-IR | 2.97 | 5.68 |
| 40-PCM | 5.9 | 7.35 |
| 40 | 1.08 | 0.9 |
| 50-PCM-IR | 1.36 | 2 |
| 50-IR | 0.44 | 1.44 |
| 50-PCM | 2.18 | 0.88 |
| 50 | 0.56 | 3.36 |
| 60-PCM-IR | 1.65 | 1.3 |
| 60-IR | 1.18 | 3.38 |
| 60-PCM | 0.5 | 4.92 |
| 60 | 0.97 | 0.35 |

temperature. Consequently, the temperature regulator enhanced the thermal environment within the dryer and effectively minimized overheating to an acceptable level under nearly all drying conditions. This suggests that the electronic damper efficiently controlled the incoming air, preventing excessive temperature increases. Furthermore, the data indicate that when the damper was not in use, the highest average temperature increase (11.13%) occurred at 40 °C with IR and PCM. In contrast, using the electronic damper under these conditions reduced the temperature difference to 1.47%. This suggests that at lower temperatures, the damper was particularly effective when combined with IR and PCM.

### Energy and exergy analysis

**SEC analysis.** Fig 12 illustrates the SEC under various drying conditions. The results show that SEC ranged from 171,048–379,601 kJ per kg of water evaporated. The highest SEC was recorded at 40°C without IR and PCM, while the lowest occurred at 60°C with both IR and PCM. However, the electricity-based SEC, which accounts only for electrical energy consumption, ranged from 20,905–64,840 kJ per kg of water evaporated. The highest value was observed at 40°C with both IR and PCM, while the lowest occurred at 60°C with PCM but without IR. In general, as temperature increases, drying time decreases, leading to lower electrical energy consumption. Consequently, SEC tends to decrease with rising temperatures. Additionally, the use of PCM alone across all temperature ranges resulted in a lower SEC, highlighting its beneficial effects. In contrast, the implementation of IR led to an increase in SEC, likely due to higher electrical energy consumption replacing solar energy when IR was activated.

In another study on the SEC of solar drying for lemon verbena leaves, results showed that SEC increased at higher temperatures [26].

**EUR analysis.** Fig 13 illustrates the average EUR during the drying process under various conditions. The results show that the average EUR ranged from 0.036 to 0.111, with the highest value recorded at 60°C using PCM and without IR exposure. In contrast, the lowest EUR was observed at 40°C with both IR and PCM. Several factors contributed significantly to the reduction in EUR:

1. Prolonged operation of the electric fan: At lower temperatures, extended drying times led to increased electrical energy consumption.

2. Activation of the IR lamp: The use of IR increased electrical energy consumption, which in turn reduced EUR.

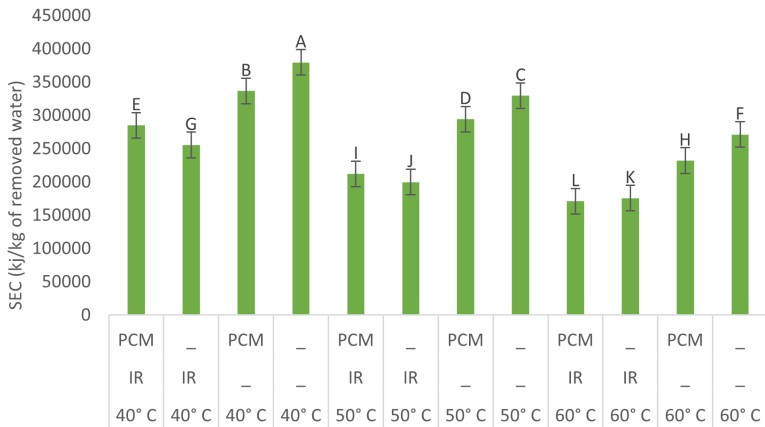

**Fig 12. Specific energy consumption in the drying of potato samples under various conditions.**

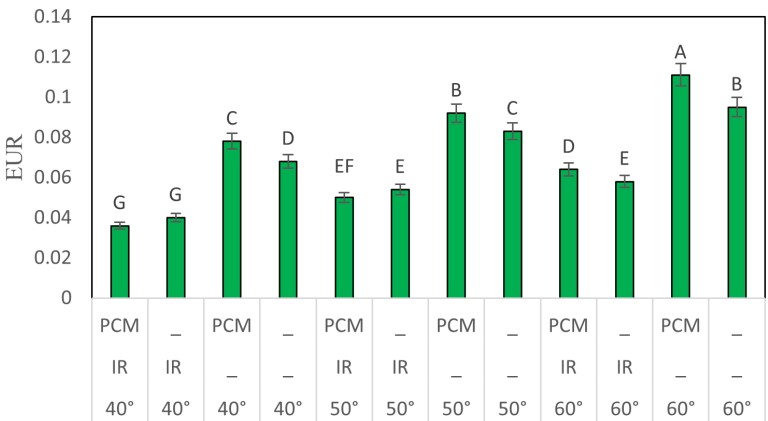

**Fig 13. EUR under various drying conditions.**

Thus, increasing the temperature while avoiding IR enhances the efficient use of solar energy, leading to lower electrical energy consumption and higher EUR. The next section will further explore the impact of various factors on EUR.

The following section provides a detailed analysis of the obtained EUR results.

The results demonstrate that an increase in temperature correlates with a rise in EUR across all experimental conditions. The average EUR values recorded at 40, 50, and 60°C were 0.055, 0.070, and 0.082, respectively. This indicates that higher temperatures promote more efficient energy utilization for drying, thereby enhancing EUR. The findings also reveal that PCM has varying effects on EUR. The average EUR for experiments incorporating PCM was 0.072, compared to 0.066 for those conducted without PCM, suggesting that PCM positively influences EUR. Additionally, the average EUR for experiments conducted with and without infrared (IR) was 0.050 and 0.088, respectively, indicating a 43% reduction in EUR when IR was used. Given that IR requires additional electrical energy consumption, it may be more beneficial to avoid its use.

**Exergy Analysis.** Fig 14 presents the exergy efficiency results for drying experiments conducted at various temperatures without PCM and IR. The findings indicate an increasing trend in average exergy efficiency, with values of 67%, 77%, and 81% at 40, 50, and 60°C, respectively. This increase can be attributed to higher drying rates at elevated

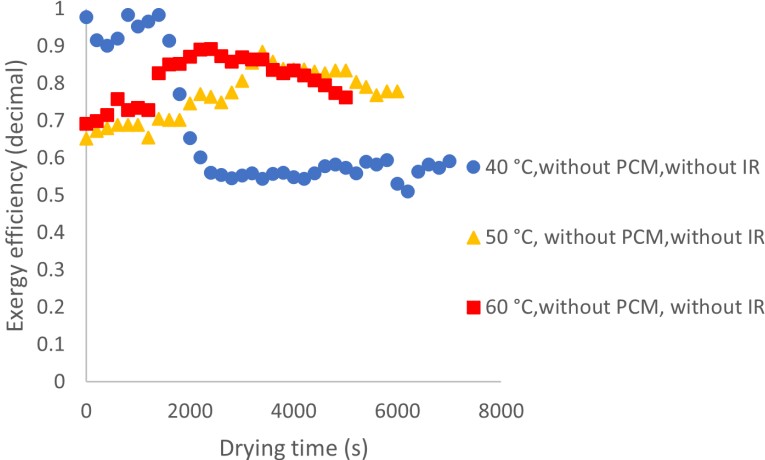

**Fig 14. Exergy efficiency at various temperatures without use of PCM and IR.**

temperatures, which accelerate the rise in exit air temperature and enhance exergy efficiency. Temperature plays a crucial role in influencing the drying kinetics and exergy efficiency of agricultural products. Higher temperatures generally improve moisture removal rates, leading to greater exergy efficiency. In a similar study calculating the exergy efficiency of cabinet drying for lemon verbena leaves at different temperatures, the results indicated that exergy efficiency increased with rising temperature [26].

Additionally, a comparison of exergy efficiency trends over drying time at different temperatures reveals that at 40 °C, the efficiency follows an irregular pattern, generally declining over time. This suggests that low drying rates lead to greater exergy absorption during the drying process. In contrast, at 50 and 60 °C, the trend differs: initially stable, then increasing, and finally declining after reaching a peak. This behavior may be attributed to the consistent removal of free moisture in the early stages, which helps stabilize exergy efficiency. As drying progresses and surface moisture decreases, the product temperature rises, causing an increase in exit air temperature and, consequently, output exergy. However, once the moisture content reaches a critical level, the incoming heat is primarily used for internal moisture migration and evaporation, leading to a decline in output exergy and exergy efficiency. Fig 15 illustrates the variations in exergy efficiency throughout the drying process under conditions with PCM and without IR. At 50 and 60 °C, the exergy efficiency trend appears more uniform compared to 40 °C. This is likely due to the cyclic absorption and release of heat by the PCM at higher temperatures, which promotes a more consistent temperature distribution within the product bed, ultimately enhancing exergy efficiency.

Fig 16 presents the exergy efficiency at various temperatures under conditions with IR and without PCM. The data show that all curves follow an upward trend, indicating a decrease in the amount of exergy absorbed by the product during drying. The average exergy efficiencies at 40, 50, and 60°C were 20%, 22%, and 33%, respectively, reflecting a reduction compared to experiments conducted without IR. Similarly, a study on solar drying of apples and mint, with and without IR, found that using IR led to a decrease in exergy efficiency [27].

Fig 17 presents the exergy efficiency values for the case where both IR and PCM are utilized. The curves display a more uniform trend compared to the scenario without PCM, likely due to PCM activation at all temperatures facilitated by the presence of IR. The average exergy efficiency at 40, 50, and 60 °C is 14%, 25%, and 29%, respectively. PCM can store and release thermal energy, helping to maintain a stable drying temperature and reduce energy consumption. Research has shown that integrating PCM with solar drying systems can significantly improve exergy efficiency by ensuring a consistent heat supply during periods of low solar radiation [28]. By absorbing excess heat during peak temperatures

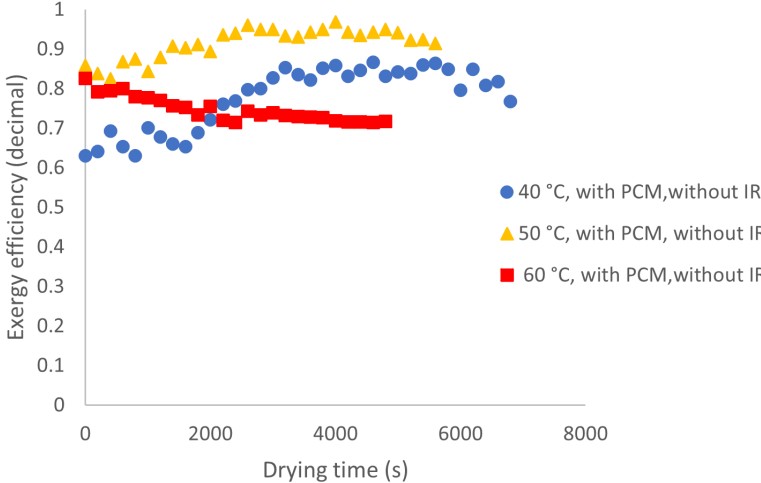

**Fig 15. Exergy efficiency across various temperatures with PCM and without IR.**

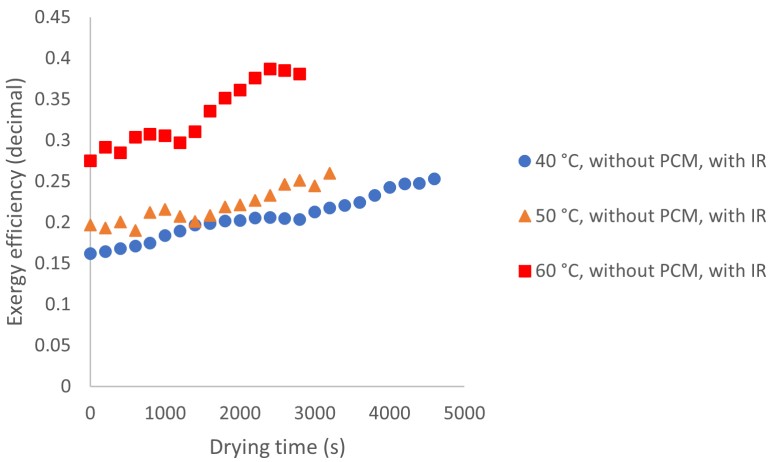

**Fig 16. Exergy efficiency at different temperatures without PCM and with IR utilization.**

and releasing it during cooler periods, PCM helps maintain optimal drying conditions, ultimately enhancing overall energy efficiency.

## Conclusion

This study successfully demonstrated the effectiveness of integrating PCM and IR technologies into a hybrid solar dryer for optimizing the drying process of potato slices. The key findings include:

1. Higher temperatures reduced drying time and improved energy and exergy efficiency, with 50°C identified as the optimal temperature for balancing drying efficiency and product color quality.

2. PCM enhanced thermal stability, reduced energy consumption, and increased exergy efficiency by storing and releasing heat during the drying process, particularly at moderate temperatures.

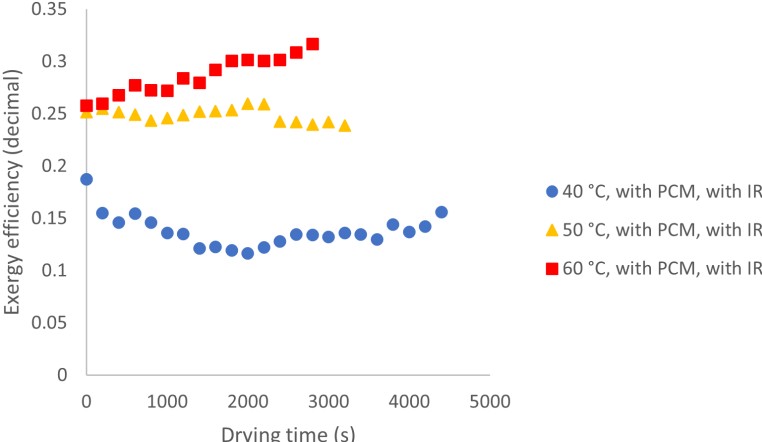

**Fig 17. Exergy efficiency across various temperatures with PCM and IR utilization.**

3. IR significantly shortened drying time but increased energy consumption, making it more suitable for time-sensitive applications rather than energy-efficient drying.

4. The combined use of PCM and IR improved thermal stability but slightly extended drying time due to PCM's heat absorption, highlighting the need for careful selection based on specific drying objectives.

The optimal drying conditions were determined to be 60°C with PCM and with IR, achieving a desirable balance between energy and exergy efficiency, drying time, and color preservation.

This study highlights the significance of advanced thermal control mechanisms in solar drying systems to improve performance, preserve product quality, and minimize energy consumption. From a practical application perspective, the research findings can be used to determine the optimal drying conditions for potatoes, balancing energy efficiency and drying time while maintaining color quality. Future research could explore the combined effects of these parameters with additional techniques, such as microwave and ultrasound-assisted drying, to further assess their impact on product quality.

## Author contributions

**Conceptualization:** Mahdi Keramat-Jahromi.

**Methodology:** Mehdi Moradi, Reza Raeesi.

**Project administration:** Mehdi Moradi.

**Software:** Mehdi Moradi, Reza Raeesi.

**Supervision:** Mehdi Moradi, Mahdi Keramat-Jahromi.

**Writing – review & editing:** Dariush Zare, Mahdi Keramat-Jahromi.

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
