## [Decision Letter · Decision Letter 0]

25 Feb 2025

Dear Dr. Moradi,

We look forward to receiving your revised manuscript.

Kind regards,

Morteza Taki, Ph.D

Academic Editor

PLOS ONE

Journal Requirements:

4. In the online submission form, you indicated that “Data may be available upon request from the journal.”

**Additional Editor Comments:**

Dear authors

I have received feedback from several experts in the field of drying, and I have also reviewed the paper myself. In my opinion, the paper requires significant revisions and should be carefully re-evaluated. Please review the comments provided and address them accordingly, so that I can conduct a thorough re-evaluation of the paper.

Sincerely

M.Taki

Reviewers' comments:

Reviewer's Responses to Questions

**Comments to the Author**

1. Is the manuscript technically sound, and do the data support the conclusions?

Reviewer #1: No

Reviewer #2: Partly

Reviewer #3: Partly

2. Has the statistical analysis been performed appropriately and rigorously?

Reviewer #1: No

Reviewer #2: No

Reviewer #3: Yes

3. Have the authors made all data underlying the findings in their manuscript fully available?

Reviewer #1: Yes

Reviewer #2: No

Reviewer #3: No

4. Is the manuscript presented in an intelligible fashion and written in standard English?

Reviewer #1: No

Reviewer #2: No

Reviewer #3: Yes

Reviewer #1: After thoroughly reviewing the research, we decided to reject it for not meeting the standards of this journal for the following reasons:

- The article is not organized.

- The current English form is not up to the journal's standards of quality.

- The introduction is very short and does not contain convincing references.

- In scientific research, innovativeness is an important criterion for evaluating research value and contributions. However, this paper does not present novel viewpoints, methods, or discoveries, thereby lacking innovativeness. The conclusions of the study also lack quantitative and in-depth descriptions and analysis.

- The references on which the research was based are insufficient.

- Absence of figure headings.

Reviewer #2: Dear authors

I think the paper need some revisions. Please check the comments:

• Clarity and Structure :

• The abstract provides a good overview of the study but could benefit from more concise language in certain areas. For instance, the sentence about the mixed effects of PCM and IR could be rephrased for better clarity.

• The introduction effectively sets the stage for the research but could include a broader review of related literature to provide context for the importance of this study within the field. The below references can improve it. I suggest to use them:

• Liu, W., Wu, Y., Bao, X., Sun, L., Xie, Y.,... Chen, Y. (2025). High-Performance Infrared Self-Powered Photodetector Based on 2D Van der Waals Heterostructures. Advanced Functional Materials, 2421525. doi: https://doi.org/10.1002/adfm.202421525

• Hao, R., Zhu, L., Shang, T., Xu, Z., & Wu, Q. (2024). Strong absorption of silica over full solar spectrum boosted by interfacial junctions and light–heat–storage of Mg(OH)2–(CrOx–SiO2). Chemical Engineering Journal, 497, 154979. doi: https://doi.org/10.1016/j.cej.2024.154979

• Jia, S., Li, Y., Gao, C., Liu, G., Ren, Y., He, C., & An, X. T. (2025). Realization of p-type MA-based perovskite solar cells based on exposure of the (002) facet. Applied Physics Letters, 126(2).

• Gao, C., Jia, S., Yin, X., Li, Z., Yang, G., Chen, J., ... & An, X. (2025). Enhancing open-circuit voltage in FAPbI 3 perovskite solar cells via self-formation of coherent buried interface FAPbI x Cl 3− x. Chemical Communications, 61(13), 2758-2761.

Materials and Methods :

• The description of the solar dryer setup is thorough, but additional diagrams or schematics might help readers better understand the system configuration.

• It would be beneficial if the authors provided more details on how the temperature controller operates and its accuracy in maintaining the desired temperatures.

Experimental Design :

• The experimental design involving three different air temperatures with varying configurations of PCM and IR is well thought out. However, it would be helpful to know why these specific temperatures were chosen and whether they represent typical conditions for potato drying.

• The inclusion of a control group without any enhancements (neither PCM nor IR) is appropriate, but the authors should discuss why this baseline condition is relevant.

Results :

• The results section is comprehensive, with clear figures and tables illustrating the data. However, some of the figures, such as those showing exergy efficiency trends, might need better labeling or annotations for easier interpretation.

• The authors should consider discussing potential outliers or unexpected findings in greater detail to enhance the robustness of their conclusions.

Discussion :

• The discussion adequately interprets the results, but it could delve deeper into the implications of the findings for practical applications in agricultural settings.

• The comparison between different drying methods (e.g., IR vs. non-IR) is insightful, but expanding on the trade-offs between energy consumption and product quality would add value.

Conclusion :

• The conclusion succinctly summarizes the key findings, particularly emphasizing the optimal drying conditions at 50°C with PCM and without IR. However, suggesting future research directions could further strengthen the paper

Reviewer #3: The paper is titled "Optimizing Solar Drying Efficiency: Effects of PCM and IR on Energy and Exergy Performance." I think this study is interesting and original. However, the paper has needed some revisions. The originality of the article is not clearly stated. Differences should be clearly stated with similar studies. If the authors fix the paper, it can be accepted. So, after the article is revised, I need to control it again. Here are some of my comments.

Attached is a PDF file containing the comments. Thank you.

**Do you want your identity to be public for this peer review?** For information about this choice, including consent withdrawal, please see our Privacy Policy

Reviewer #1: No

Reviewer #2: No

Reviewer #3: No

---

## [Author Response · Author response to Decision Letter 1]

5 Apr 2025

In the name of God,

Dear Dr. Taki,

I sincerely appreciate your time and consideration of my manuscript. We sincerely appreciate you and the esteemed reviewers for your careful consideration of the manuscript and your valuable comments. We have carefully revised the manuscript and addressed all the reviewers' comments. Thank you for your valuable feedback.

Reviewers' comments:

Comments to the Author

1. Is the manuscript technically sound, and do the data support the conclusions?

Reviewer #1: No

Reviewer #2: Partly

Reviewer #3: Partly

Response: The manuscript has been thoroughly revised for clarity and quality. However, any additional comments or suggestions are greatly appreciated.________________________________________

2. Has the statistical analysis been performed appropriately and rigorously?

Reviewer #1: No

Reviewer #2: No

Reviewer #3: Yes

Response: Thank you. The statistical analysis was performed using SPSS 16.0, and this is explained in the manuscript. If there are any questions regarding ambiguity, I would be happy to address them.

3. Have the authors made all data underlying the findings in their manuscript fully available?

Reviewer #1: Yes

Reviewer #2: No

Reviewer #3: No

Response: All data used in the manuscript is presented in the form of curves or tables. However, if the raw data is required, it can be provided upon request.________________________________________

4. Is the manuscript presented in an intelligible fashion and written in standard English?

Reviewer #1: No

Reviewer #2: No

Reviewer #3: Yes

Response: The manuscript has been completely revised. ________________________________________

5. Review Comments to the Author

Reviewer #1:

After thoroughly reviewing the research, we decided to reject it for not meeting the standards of this journal for the following reasons:

- The article is not organized.

- The current English form is not up to the journal's standards of quality.

- The introduction is very short and does not contain convincing references.

- In scientific research, innovativeness is an important criterion for evaluating research value and contributions. However, this paper does not present novel viewpoints, methods, or discoveries, thereby lacking innovativeness. The conclusions of the study also lack quantitative and in-depth descriptions and analysis.

- The references on which the research was based are insufficient.

- Absence of figure headings.

Response: The manuscript has been thoroughly revised, and the number of references has been increased. Additionally, the titles of the figures, which were also present in the previous version of the manuscript, are included.

Reviewer #2: Dear authors

I think the paper need some revisions. Please check the comments:

• Clarity and Structure :

• The abstract provides a good overview of the study but could benefit from more concise language in certain areas. For instance, the sentence about the mixed effects of PCM and IR could be rephrased for better clarity.

• The introduction effectively sets the stage for the research but could include a broader review of related literature to provide context for the importance of this study within the field. The below references can improve it. I suggest to use them:

• Liu, W., Wu, Y., Bao, X., Sun, L., Xie, Y.,... Chen, Y. (2025). High-Performance Infrared Self-Powered Photodetector Based on 2D Van der Waals Heterostructures. Advanced Functional Materials, 2421525. doi: https://doi.org/10.1002/adfm.202421525

• Hao, R., Zhu, L., Shang, T., Xu, Z., & Wu, Q. (2024). Strong absorption of silica over full solar spectrum boosted by interfacial junctions and light–heat–storage of Mg(OH)2–(CrOx–SiO2). Chemical Engineering Journal, 497, 154979. doi: https://doi.org/10.1016/j.cej.2024.154979

• Jia, S., Li, Y., Gao, C., Liu, G., Ren, Y., He, C., & An, X. T. (2025). Realization of p-type MA-based perovskite solar cells based on exposure of the (002) facet. Applied Physics Letters, 126(2).

• Gao, C., Jia, S., Yin, X., Li, Z., Yang, G., Chen, J., ... & An, X. (2025). Enhancing open-circuit voltage in FAPbI 3 perovskite solar cells via self-formation of coherent buried interface FAPbI x Cl 3− x. Chemical Communications, 61(13), 2758-2761.

Response: Thank you. All comments provided by Reviewer #2 have been addressed. The abstract has been revised, and the introduction has been expanded with additional references, including those suggested by the reviewer.

Materials and Methods :

• The description of the solar dryer setup is thorough, but additional diagrams or schematics might help readers better understand the system configuration.

Response: Thank you. Schematic Fig. (4) explains the solar dryer. However, any other areas of ambiguity can be addressed upon the reviewer’s request.

• It would be beneficial if the authors provided more details on how the temperature controller operates and its accuracy in maintaining the desired temperatures.

Response: Some additional explanations have been added to the temperature controller section. Regarding temperature setting accuracy, the following sentence has been included: "An SSR model 25DA (produced by CRYDOM) and a PT100 temperature sensor with an accuracy of ±0.1 °C were installed at the output of the drying chamber."

Experimental Design :

• The experimental design involving three different air temperatures with varying configurations of PCM and IR is well thought out. However, it would be helpful to know why these specific temperatures were chosen and whether they represent typical conditions for potato drying.

Response: Thank you. The working temperatures were selected based on the literature, which has been addressed in the revised manuscript.

• The inclusion of a control group without any enhancements (neither PCM nor IR) is appropriate, but the authors should discuss why this baseline condition is relevant.

Response: Thank you. To investigate the effects of PCM and IR, it is essential to include conditions without them as well. In other words, the research hypothesis assumes that PCM and IR positively impact potato drying; therefore, we also consider scenarios without PCM and IR for comparison.

Results :

• The results section is comprehensive, with clear figures and tables illustrating the data. However, some of the figures, such as those showing exergy efficiency trends, might need better labeling or annotations for easier interpretation.

• The authors should consider discussing potential outliers or unexpected findings in greater detail to enhance the robustness of their conclusions.

Response: Thank you. We have thoroughly revised the manuscript and relabeled the figures to improve clarity and ease of interpretation.

Discussion :

• The discussion adequately interprets the results, but it could delve deeper into the implications of the findings for practical applications in agricultural settings.

• The comparison between different drying methods (e.g., IR vs. non-IR) is insightful, but expanding on the trade-offs between energy consumption and product quality would add value.

Response: Thank you for your comment. The conclusion has been revised accordingly.

Conclusion :

• The conclusion succinctly summarizes the key findings, particularly emphasizing the optimal drying conditions at 50°C with PCM and without IR. However, suggesting future research directions could further strengthen the paper.

Response: Thank you for your feedback. The manuscript has been thoroughly revised. However, due to adjustments in the SEC analysis, the optimal drying condition is now determined to be 60°C with PCM and IR. Additionally, regarding future research, we have included the following sentence:

"Future research could explore the combined effects of these parameters with additional techniques, such as microwave and ultrasound-assisted drying, to further assess their impact on product quality."

Reviewer #3

Sincerely, Dear editor-in-chief, Dr. Morteza Taki The paper is titled "Optimizing Solar Drying Efficiency: Effects of PCM and IR on Energy and Exergy Performance." I think this study is interesting and original. However, the paper has needed some revisions. The originality of the article is not clearly stated. Differences should be clearly stated with similar studies. If the authors fix the paper, it can be accepted. So, after the article is revised, I need to control it again. Here are some of my comments.

1.The abstract could be expanded to include more detailed information.

Response: Thanks. Abstract has been revised.

2. In the introduction, the authors should connect the state of the art to the goals of their paper by providing a clear and concise analysis. This analysis should identify existing knowledge gaps and relate them to the paper's objectives. Additionally, the authors need to explain the novelty and relevance of their goals. There are numerous relevant papers that should be cited and discussed in this section. However, few studies and articles focusing on the energy-exergy analysis of dryers have been investigated.

Response: The introduction has been revised, and additional relevant studies have been investigated.

3.Enhance the image quality; Figures 2 and 5 can be improved further.

Response: Thank you. Figures have been provided in TIF format according to the journal guidelines.

4. The authors should enhance their discussion of the results and compare their findings with those of other relevant research studies.

Response: some other related references added into the manuscript.

5. The authors need to include a more comprehensive discussion about the novelty of their paper.

Response: Thank you. The novelty of this study lies in investigating the combined effects of IR, temperature, and PCM on drying kinetics, as well as energy and exergy efficiency. This has been highlighted in the manuscript. Any further suggestions are welcome.

6. Discuss the potential of using this technology in that area in the future.

Response: Thank you. The conclusion section has been revised accordingly.

7. Incorporate recent studies on drying potatoes and compare the findings of this research with those sources. Addressing these suggestions could substantially enhance the manuscript, making it more suitable for publication in the journal.

Response: Thank you. Relevant research on potato drying has been added to the manuscript.

8. A graphical abstract could be included.

Response: Thanks. It was provided and added to the end of the manuscript.

9. Here are some tips and questions to consider:

1) Throughout the text of the manuscript, the term product color quality should replace product quality because the latter includes several factors, and the color change is only one of them.

Response: Thank you. The necessary replacements have been made throughout the manuscript.

2) The author stated that a parabolic collector was used, although it appears to have been implemented as a flat type (Generally, the parabolic design requires a tracking system).

Response: Thank you. The employed collector was a concentrated parabolic collector with a fixed angle relative to the horizon. This collector has been used in previous research, which has been cited in the manuscript (Ebadi et al., 2021). (Ebadi, H., Zare, D., Ahmadi, M., & Chen, G. (2021). Performance of a hybrid compound parabolic concentrator solar dryer for tomato slices drying. Solar Energy, 215, 44-63.‏ https://doi.org/10.1016/j.solener.2020.12.026)

3) To control the temperature of the dryer, why use a damper rather than a variablespeed fan? A higher mass flow also reduces drying time.

Response: Thank you. As explained in the manuscript, overheating of the drying air was an issue with the collector. To address this problem, a damper was installed at the collector's output, as shown in Figure 3, allowing ambient air to mix with the heated air and regulate its temperature.

Fig 3. Air temperature regulator.

4) The specific energy consumption was calculated using equation 3, excluding the energy received by the solar collector. This aspect should be revised and recalculated.

Response: Thank you. The manuscript has been revised, and solar energy has been included in the SEC computation.

5) Some parameters, such as SEC and EUR, are not mentioned in the text.

Response: Thank you. These terms —Specific Energy Consumption (SEC) and Energy Utilization Ratio (EUR)—are fully defined in both the abstract and the introduction.

6) Equation 5 does not consider the specific heat of the product or the energy needed to raise its temperature.

Response: Thank you. Based on the definition, useful energy refers to the power consumed for evaporating moisture from the drying material. This definition has been introduced before Equation (5). Similar studies have also used Equation (5) for useful energy calculations, as cited in the manuscript ([Yogendrasasidhar & Setty, 2018]; [Moradi et al., 2020]):

Yogendrasasidhar, D., & Setty, Y. P. (2018). Drying kinetics, exergy and energy analyses of Kodo millet grains and Fenugreek seeds using wall heated fluidized bed dryer. Energy, 151, 799–811.

Moradi, M., Fallahi, M. A., & Mousavi Khaneghah, A. (2020). Kinetics and mathematical modeling of thin layer drying of mint leaves by a hot water recirculating solar dryer. Journal of Food Process Engineering, 43(1), e13181.‏ https://doi.org/10.1111/jfpe.13181

7) In the exergy calculation section, how did you calculate the exergy of the PCM?

Response: Thank you. Since PCM is located beneath the product tray, its presence alters the outlet air temperature of the drying chamber. In other words, the presence or absence of PCM affectsTout in Equation (7).

8) Reference No. 15 in line 299 of the examined product is strawberries, not apple slices. Additionally, the mint variety (Mentha spicata L.) used in reference number 16 (line 301) is written incorrectly as Mentha piperita L.

Response: Thank you. You are correct. The necessary corrections have been made.

9) The implementation of Phase Change Materials (PCM) is often driven by the need for energy storage and its utilization during periods when solar energy is unavailable. This aspect appears to have been overlooked in this research.

Response: Thank you for your comment. This study presents a practical application of PCM in solar dryers, which could be beneficial when solar energy is unavailable. However, our primary objective is to investigate the effect of PCM, in combination with other drying factors (temperature and IR), on the energy and exergy efficiency of potato drying. This idea could be explored further in future research.

---

## [Decision Letter · Decision Letter 1]

6 May 2025

Optimizing Solar Drying Efficiency: Effects of PCM, and IR on Energy and Exergy Performance

PONE-D-25-06470R1

Dear Dr. Moradi,

We’re pleased to inform you that your manuscript has been judged scientifically suitable for publication and will be formally accepted for publication once it meets all outstanding technical requirements.

Kind regards,

Morteza Taki, Ph.D

Academic Editor

PLOS ONE

Additional Editor Comments (optional):

Dear authors

I am pleased to inform you that all the reviewers have approved your revisions, and I have personally evaluated the manuscript and consider it suitable for publication.

Best Regards

M.Taki

Reviewers' comments:

Reviewer's Responses to Questions

**Comments to the Author**

Reviewer #2: All comments have been addressed

Reviewer #3: All comments have been addressed

2. Is the manuscript technically sound, and do the data support the conclusions?

Reviewer #2: Yes

Reviewer #3: Yes

3. Has the statistical analysis been performed appropriately and rigorously?

Reviewer #2: Yes

Reviewer #3: Yes

4. Have the authors made all data underlying the findings in their manuscript fully available?

Reviewer #2: Yes

Reviewer #3: Yes

5. Is the manuscript presented in an intelligible fashion and written in standard English?

Reviewer #2: Yes

Reviewer #3: Yes

Reviewer #2: Dear Authors

I evaluated the revised paper and I think the paper is ready for publication and all the comments were addressed.

Reviewer #3: I appreciate the authors' efforts in addressing my comments to the best of their ability, which has significantly improved the quality of the manuscript.

**Do you want your identity to be public for this peer review?** For information about this choice, including consent withdrawal, please see our Privacy Policy

Reviewer #2: No

Reviewer #3: No

---

## [Editor Report · Acceptance letter]

PONE-D-25-06470R1

PLOS ONE

Dear Dr. Moradi,

I'm pleased to inform you that your manuscript has been deemed suitable for publication in PLOS ONE. Congratulations! Your manuscript is now being handed over to our production team.

Kind regards,

on behalf of

Dr. Morteza Taki

Academic Editor

PLOS ONE